

# Colour-emotion associations in individuals with red-green colour blindness

Domicele Jonauskaite[1], Lucia Camenzind[1], C. Alejandro Parraga[2], Cécile N. Diouf[1], Mathieu Mercapide Ducommun[1], Lauriane Müller[1], Mélanie Norberg[1] and Christine Mohr[1]

[1] Institute of Psychology, University of Lausanne, Lausanne, Vaud, Switzerland
[2] Comp. Vision Centre/Comp. Sci. Department, Universitat Autònoma de Barcelona, Barcelona, Spain

## ABSTRACT

Colours and emotions are associated in languages and traditions. Some of us may convey sadness by saying *feeling blue* or by wearing black clothes at funerals. The first example is a conceptual experience of colour and the second example is an immediate perceptual experience of colour. To investigate whether one or the other type of experience more strongly drives colour-emotion associations, we tested 64 congenitally red-green colour-blind men and 66 non-colour-blind men. All participants associated 12 colours, presented as terms or patches, with 20 emotion concepts, and rated intensities of the associated emotions. We found that colour-blind and non-colour-blind men associated similar emotions with colours, irrespective of whether colours were conveyed via terms ($r = .82$) or patches ($r = .80$). The colour-emotion associations and the emotion intensities were not modulated by participants' severity of colour blindness. Hinting at some additional, although minor, role of actual colour perception, the consistencies in associations for colour terms and patches were higher in non-colour-blind than colour-blind men. Together, these results suggest that colour-emotion associations in adults do not require immediate perceptual colour experiences, as conceptual experiences are sufficient.

## INTRODUCTION

We *feel blue*, *see red*, and have some *black days*. As Westerners, we might wear *white* to weddings and *black* to funerals. These examples show that colours and affective meanings are associated in natural languages and cultural traditions. Moreover, colour-emotion associations are highly similar across cultures (*Adams & Osgood, 1973*; *D'Andrade & Egan, 1974*; *Gao et al., 2007*; *Ou et al., 2018*; *Jonauskaite et al., 2020a*). This similarity is indicative of a human psychological universal, which is a mental attribute shared by all or nearly all healthy human beings (see *Norenzayan & Heine, 2005*). This universal might be determined by conceptual knowledge, because emotion associations were similar when matched to colour patches or colour words, indicating that immediate colour perception is not necessary for these associations to be reported (*Jonauskaite et al., 2020b*). To test this possibility, we recruited individuals with congenital red-green colour blindness. Such

Corresponding author
Domicele Jonauskaite,
domicele.jonauskaite@unil.ch

individuals have never seen colours in the same way as individuals with intact colour vision due to their congenital deficiencies (*Linhares, Pinto & Nascimento, 2008*). Yet, colour-blind individuals have been exposed to similar conceptual information, namely similar cultural and linguistic environments as non-colour-blind individuals (*Byrne & Hilbert, 2010*). If colour-emotion associations in the two groups are similar, irrespective of whether seeing colour patches or colour terms, we would have good reasons to conclude that colour-emotion associations are so well established that conceptual processing is sufficient, and that immediate colour perception is not essential for such associations to be reported.

We are aware of several older and more recent studies investigating the extent to which colour-emotion associations are shared across cultures (*Adams & Osgood, 1973*; *D'Andrade & Egan, 1974*; *Hupka et al., 1997*; *Madden, Hewett & Roth, 2000*; *Gao et al., 2007*; *Ou et al., 2018*; *Jonauskaite et al., 2020a*). Some studies reported cross-cultural similarities in, and even claimed universality for, associations between brighter colours and positivity (*Specker et al., 2018*), associations between colours and affective dimensions (*Adams & Osgood, 1973*; *Gao et al., 2007*; *Ou et al., 2018*), or colours and emotion terms (*D'Andrade & Egan, 1974*). Examples of these associations include *red* being an active, warm and strong colour, *blue*, *green*, and *white* being positive colours, dark colours being heavy while desaturated colours being passive. Other studies highlighted cross-cultural differences. For instance, *envy* was associated with *black*, *red*, *green*, *yellow*, or *purple* depending on the nation (*Hupka et al., 1997*). These studies, however, used different methods, usually testing a limited number of colours, emotions, and/or cultures.

Recently, Jonauskaite and colleagues (*Jonauskaite et al., 2020a*) tested 240 colour-emotion associations in 30 nations resulting from associations between 12 colour terms and 20 emotion concepts. For each colour term, participants were free to associate as many emotions as they felt appropriate, in their native language. Results revealed high similarity in the way colours and emotions were associated across nations (average correlation was $r = .88$). These cross-cultural results indicate that humans largely share how they associate colours with emotions, at least when colours are presented as terms. Presentation mode does not seem to matter, though, as similar emotions have been associated with colour patches, at least when participants were tested in a Western context. More precisely, in another study, Swiss adults again associated the 12 colours with the 20 emotion concepts (*Jonauskaite et al., 2020b*). One group of participants associated emotions with basic colour terms and the other group with focal colours that best represent these basic colour terms. Both groups chose similar emotions for the same colour concepts, irrespective of whether they were presented as terms or patches (correlation between groups was $r = .82$). In a different study, Wang and colleagues (*2014*) reported high similarity in term-patch associations for *blue* but not *red*. In their study, Chinese participants evaluated r*ed* more positively as a term than a patch. Overall, with some potential exceptions, these results suggest that seeing a colour is not key to decide on colour-emotion associations, at least once individuals have reached adulthood.

High similarities in colour-emotion associations across cultures and presentation mode do not reveal the mechanisms that drive the formation of shared colour-emotion associations. Considering potential mechanisms, one mechanism might be shared

perceptual experiences by most humans (see also, *Palmer & Schloss, 2010*; *Jonauskaite et al., 2019a*). In this case, a direct perceptual experience of colour might lead to an affective experience. For instance, looking at a colour would make one *feel* a certain emotion or immediately remind of a particular emotion. Another mechanism might be shared conceptual knowledge, accessed and transmitted through language (see *Xu, Dowman & Griffiths, 2013*, for cultural transmission of colour terms). In that case, colours and emotions would be conceptually associated without necessarily meaning that immediate colour perception itself evokes associations with affective experiences.

To test the relative importance of conceptual versus immediate colour experience, we suggest for this study to test colour-emotion associations in populations with colour vision deficiencies. The most frequent colour vision deficiency is congenital red-green colour blindness. Here, affected individuals can discern a smaller number of colours than individuals with complete colour vision (*Neitz & Neitz, 2000*; *Linhares, Pinto & Nascimento, 2008*). Red-green colour blindness, also called Daltonism after John Dalton (*Dalton, 1798*), affects around 8% of the male population and around 0.6% of the female population of European-Caucasian origin (*Sharpe et al., 1999*; *Birch, 2012*). Such individuals confuse certain colours along the red-green axis (e.g., *red* and *brown*, *green* and *brown*, *pink* and *grey*, *grey* and *green,* etc.; *Moreira et al., 2014*) and likely see the world in bluish-yellowish colours (*Judd, 1949*; *Byrne & Hilbert, 2010*). Individuals with red-green colour blindness have never seen certain colours the way individuals with intact colour vision do, but have been exposed to their shared cultural and linguistic environments (e.g., traffic colours; *Almustanyir & Hovis, 2020*). Accordingly, if individuals with and without red-green colour blindness display similar colour-emotion associations, we can argue that shared conceptual knowledge is sufficient for colour-emotion associations to be reported.

Studies assessing colour naming and colour arrangements support the importance of conceptual knowledge. In case of colour naming, colour-blind individuals were able to name colours indicating that they learned to differentiate colours, irrespective of whether they look the same or different to colours perceived by individuals with intact colour vision (*Jameson & Hurvich, 1978*; *Paramei, 1996*; *Bonnardel, 2006*; *Nagy & Ábrahám, 2014*; *Moreira et al., 2014*). *Bonnardel (2006)* found that consensus in colour naming ranged between 52% and 74% for colour-blind and non-colour-blind individuals. The highest consensus emerged when participants had to name colour chips using one of eight colour terms (i.e., constrained colour naming task; 74% consensus). Some of the chips were focal colours (i.e., the best examples of each colour category) while others were not. The lowest consensus emerged for a task that least involved language (i.e., freely grouping colour chips into colour categories, 52% consensus). For colour arrangements, colour-blind individuals mentally arranged colours more similarly to non-colour-blind individuals when colours were presented as terms than as patches (*Shepard & Cooper, 1992*; *Saysani, Corballis & Corballis, 2018a*). When presented with terms, colour-blind individuals used three colour axes (i.e., red-green, blue-yellow, and dark-light) to arrange colours. When presented with patches of focal colours, colour-blind individuals collapsed colours along the red-green axis and used only two axes to arrange colours. Taken together, conceptual knowledge

seems essential for colour naming and colour arrangements. Nonetheless, colour terms and colour patches might be treated somewhat differently by colour-blind individuals. If so, colour-blind individuals might also treat colour-emotion associations differently when actually reading a colour term or seeing a colour patch.

To test the importance of conceptual knowledge and immediate perceptual colour experience, we assessed 240 colour-emotion associations in individuals with and without red-green colour blindness using a previously established methodology (*Jonauskaite et al., 2019c*; *Jonauskaite et al., 2020b*; *Jonauskaite et al., 2019a*). Part of each group associated emotion terms with colour terms while the remainder associated emotion terms with colour patches displaying focal colours. Emotion terms were presented in a circular format (*Scherer, 2005*; *Scherer et al., 2013*). We compared colour-emotion associations between colour-blind and non-colour-blind individuals as well as between colour terms and colour patches in each group.

If shared conceptual knowledge is sufficient for colour-emotion associations to be reported, we would expect high similarities in colour-emotion associations between individuals with and without colour blindness. We would also expect high similarity in colour-emotion associations between colour terms and colour patches in colour-blind individuals. If, however, previous or immediate perceptual colour experiences are necessary for consistent colour-emotion associations to be reported, we would expect differences in colour-emotion associations between individuals with and without colour blindness (e.g., see *Álvaro et al., 2015*, for colour preferences). These differences should be more pronounced when actual perceptual colours (i.e., colour patches) rather than colour terms are evaluated, since perceptual colours appear differently to individuals with and without colour blindness (*Byrne & Hilbert, 2010*). We would also expect lower consistency between colour terms and colour patches in colour-blind individuals.

In addition to comparing colour-blind and non-colour-blind individuals, we further modelled colour blindness as a continuum. We tested whether the strength of colour blindness predicted colour-emotion associations. We chose to treat colour blindness as a continuum due to variations in physiological and behavioural expressions of colour blindness. Red-green colour blindness results from changes in the photopigments in the cone receptors coding for long ("reddish"; L-cones) or medium ("greenish"; M-cones) wavelengths (*Parry, 2015*). For some individuals, cones are completely missing (dichromatic vision), while for others, they are malfunctioning (anomalous trichromatic vision). The degree of perceptual confusion is related to the degree of individuals' physiological impairments (*Neitz & Neitz, 2000*). Many previous studies considered only individuals with dichromatic vision (*Jameson & Hurvich, 1978*; *Shepard & Cooper, 1992*; *Paramei, Bimler & Cavonius, 1998*; *Moreira et al., 2014*; *Saysani, Corballis & Corballis, 2018a*). However, such individuals comprise just 28.5% of all colour-blind men of European origin (i.e., 2.3% of the general population of European males; *Sharpe et al., 1999*). Thus, we decided to freely sample from the colour-blind population and include both individuals with dichromatic vision and anomalous trichromatic vision (similar to *Paramei, 1996*; *Bonnardel, 2006*; *Nagy & Ábrahám, 2014*).

## MATERIALS & METHODS

### Participants

We recruited 130 men, 64 were colour-blind by self-report, which was confirmed with colour vision tests (see the *Colour vision tests* section for further details). About half of the participants took part in the colour terms condition (associating terms with emotions, Table 1) and the other half took part in the colour patches condition (associating patches with emotions, Table 1). All participants lived in Switzerland. Most participants were either students or staff members of a local university. They were fluent French speakers, apart from one participant who was excluded from the analyses (see Table 1). Age did not differ between study groups, $F(3, 125) = 1.50$, $p = .218$.

Based on a related previous publication, where we ran a $2 \times 12$ mixed-design MANOVA to compare emotion associations between terms and patches (*Jonauskaite et al., 2020b*), we expected a large effect size ($V = .55$). We entered this effect size in the G*Power sample size calculator (*Faul et al., 2007*) together with the expected alpha (.05) and beta (.80) levels. We obtained 26 participants as a minimal total sample size. Yet, we decided to collect more participants to be able to also consider the variability in the expression of colour blindness as well as exclude weak colour-blind participants from part of the analyses.

Participation was voluntary and remunerated with monetary reward (CHF 20 in gift vouchers). The study was conducted in accordance with the principles expressed in the Declaration of Helsinki (*World Medical Association, 2013*). We received ethics approval from the Research Ethics Commission of the University of Lausanne (C_SSP_032020_00003).

### Colour stimuli

We used *red, orange, yellow, green, turquoise, blue, purple, pink, brown, white, grey,* and *black* as colour stimuli. Eleven of these colour stimuli represent the principal colour categories (*Biggam, 2012*). We also included *turquoise* because it covers the blue–green range. In the terms condition, colour stimuli were presented as French colour terms written in black ink (*Spence, 1989*, see Table S1). In the patches condition, colour stimuli were presented as colour patches. Colour patches displayed the best exemplars of each colour category (i.e., focal colours, Table 2, *Lindsey & Brown, 2014*), and have been used in native French speakers in Switzerland (*Jonauskaite et al., 2020b*).

### Emotion assessment

We used the Geneva Emotion Wheel (GEW 3.0; Fig. 1; *Scherer, 2005*; *Scherer et al., 2013*) to measure emotion associations with colours. GEW is a validated self-report measure of the feeling component of emotion. Twenty emotion concepts are represented along the circumference of a wheel. These emotion concepts are organized along two axes. The horizontal axis represents *valence,* also known as evaluation or pleasantness (positive vs. negative). The vertical axis represents *power*, also known as control, dominance, or potency (strong vs. weak). Emotion concepts can further be categorised in terms of *arousal,* also known as activation (high arousal vs. low arousal), based on complementary research studies (*Fontaine, 2013*; *Soriano et al., 2013*). We reported this categorisation in a previous

**Table 1  Demographic information of colour-blind and non-colour-blind participants, shown by condition.**

|  |  | N | Age | | Gender | French fluency (max 8) | |
|---|---|---|---|---|---|---|---|
|  |  |  | *Mean* | *SD* |  | *Mean* | *SD* |
| Colour terms condition | Colour blind | 30 | 24.93 | 4.46 | All males | 8 | 0.00 |
|  | Non-colour-blind | 31 | 23.55 | 3.38 | All males | 8 | 0.00 |
| Colour patches condition | Colour blind | 34 | 22.56 | 5.71 | All males | 7.88 | 0.54 |
|  | Non-colour-blind | 34 | 23.53 | 3.95 | All males | 7.75 | 0.65 |

**Table 2  Colour stimuli used in the terms and patches conditions.** Munsell values for colour patches taken from *Lindsey & Brown (2014)*. The last columns show the CIE1931 xyY values for our patches.

| Colour term | Colour patch | | | | | |
|---|---|---|---|---|---|---|
|  | Munsell colour-order system | | | CIE1931 coordinates | | |
|  | Hue | Value | Chroma | Y (cd/m$^2$) | x | y |
| Red | 5.00 R | 4 | 14 | 12.00 | .57 | .31 |
| Orange | 5.00 YR | 6 | 12 | 30.05 | .51 | .42 |
| Yellow | 5.00 Y | 8 | 14 | 59.44 | .45 | .48 |
| Green | 2.50 G | 5 | 12 | 20.99 | .27 | .50 |
| Turquoise | 7.50 BG | 6 | 8 | 30.38 | .22 | .33 |
| Blue | 10.00 B | 6 | 10 | 30.05 | .20 | .24 |
| Purple | 7.50 P | 4 | 10 | 12.00 | .31 | .22 |
| Pink | 7.50 RP | 7 | 8 | 43.07 | .37 | .31 |
| Brown | 7.50 YR | 3 | 6 | 6.55 | .49 | .42 |
| White | 10.00 RP | 9.5 | 0 | 90.01 | .31 | .33 |
| Grey | 10.00 RP | 6 | 0 | 30.05 | .31 | .33 |
| Black | 10.00 RP | 1.5 | 0 | 2.02 | .31 | .33 |
| Grey (background) | 10.00 RP | 5 | 0 | 18.58 | .31 | .32 |

related study (*Jonauskaite et al., 2020b*) and here in Table S2. Circles of increasing size connect the centre of the wheel with the circumference of the wheel. These circles denote five degrees of emotion intensity, coded from 1 (smallest circle; weakest intensity) to 5 (biggest circle; strongest intensity), or 0 if no emotion is chosen (little square). The Swiss Centre for Affective Sciences provides the validated French version of the GEW (Table S1).

## Colour vision tests

Red-green colour blindness varies in severity. This variation can be behaviourally captured with colour vision tests. In this study, we used the Ishihara test (*Ishihara, 2000*), the Farnsworth test (*Farnsworth, 1947*), and the Lanthony test (*Lanthony, 1978a*; *Lanthony, 1978b*). Detailed information regarding testing and scoring of the three behavioural tests appears in Supplementary material. These and other similar behavioural tests do not seem to rely on higher cognitive functions. Rather, they rely on the discrimination of primary visual features, since they have been successfully used to assess colour vision in other animal species (e.g., dogs, seals; *Scholtyssek, Kelber & Dehnhardt, 2014*; *Siniscalchi et al., 2017*).

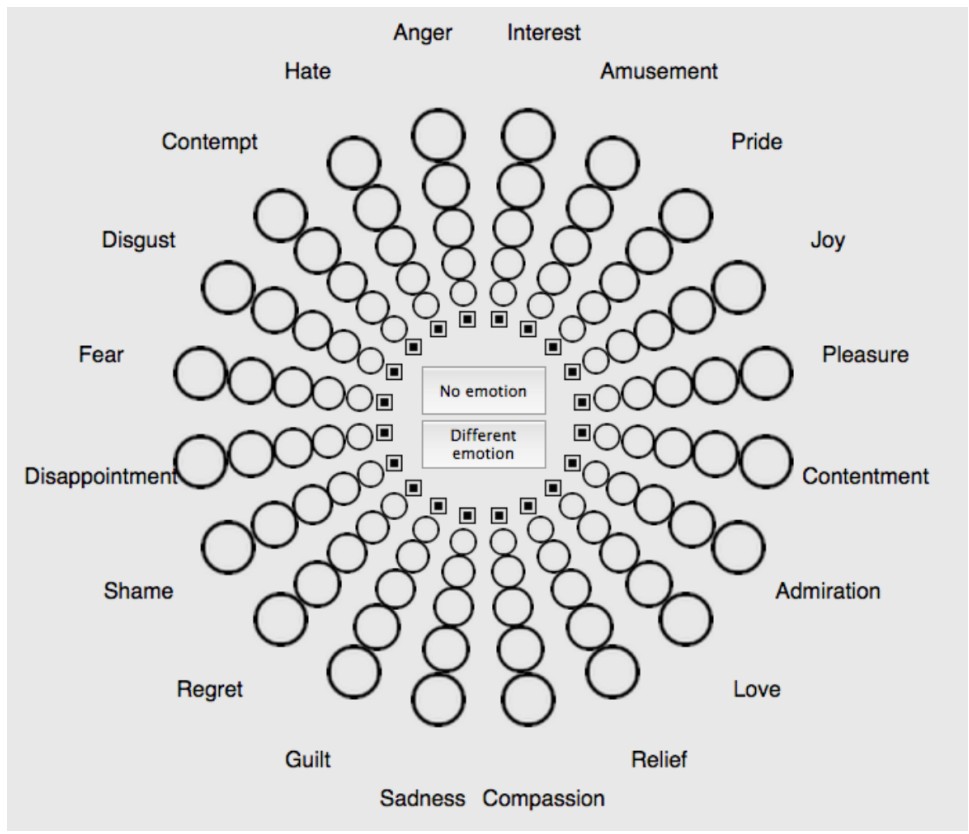

**Figure 1  Geneva Emotion Wheel (GEW) to assess colour-emotion associations with colour terms and colour patches.** The intensity of the associated emotions was assessed with circles of increasing size, smaller circles indicated less intense emotions. See Table S1 for the French version (*Scherer, 2005*; *Scherer et al., 2013*).

## Procedure

We performed the colour terms and colour patches conditions as similarly as possible, but had to also account for the different study material. The procedure was identical to a previous study (*Jonauskaite et al., 2020b*). Below, we detail what was comparable for conditions (see *Common to both conditions*), followed by the description of the terms condition procedure and the patches condition procedure.

### *Common to both conditions*

Upon arrival to the welcome room, we gave participants relevant study information. Those who agreed to participate signed the written informed consent form (see Fig. 2 for procedure). Next, we tested participants' colour vision with the Ishihara test. All colour vision tests were conducted as physical tests under the same conditions of artificial office light. Afterwards, participants were invited to the testing room. The computer monitor was the only source of illumination in the testing room. All participants performed the experiment on the same monitor: Eizo ColourEdge CG247 24.1" (inches) LCD display, with an in-built self-calibration sensor. We set the temperature of the monitors to 6500

**Figure 2** **Procedure for the colour terms and colour patches conditions.** (A) Participants received written study information and signed informed consent. (B) Participants completed the Ishihara test. (C) Main experiment. In the terms condition, participants saw 12 colour terms in randomised order. They associated colour terms with one, several, or none of the Geneva Emotion Wheel (GEW) emotion concepts (see *Emotion assessment* and Fig. 1 for enlarged GEW). In the patches condition, participants saw 12 colour patches in randomised order. They associated colour patches with one, several, or none of the GEW emotion concepts on the subsequent screen. Here, they saw the small GEW squares as well as the GEW rays of chosen emotion concepts presented in the colour they were currently evaluating. In both conditions, participants answered demographic questions. (D) In the patches condition, most participants also performed a colour-naming task. (E) Participants completed the Farnsworth D-15 and Lanthony D-15 tests in random order. (F) Participants were debriefed.

K, gamma: 2.2, contrast: 100%, and brightness: 120cd/m$^2$. Resolution was $1{,}920 \times 1{,}200$ pixels and the frame rate was 59.90 Hz. The eye-screen distance was approximately 70 cm.

Participants completed either the terms or the patches condition. Experimenters were available for questions at any point during the experiments. After the main experiment, participants returned to the welcome room and completed the Farnsworth D-15 and Lanthony D-15 tests. These tests were given in a randomised order across participants. Once participants completed the first test, the completed test was hidden and they were asked to complete the second test. Upon the completion of both tests, participants were debriefed and remunerated. Participants were invited to ask questions and received a debriefing sheet with written information and contact details for future references. The entire experiment took between 50 and 70 min.

### Colour terms condition

The colour terms condition was performed in the laboratory testing room. We used an existing online survey link (https://www2.unil.ch/onlinepsylab/colour/main.php); also used to collect data remotely for a larger ongoing International Colour-Emotion Survey online (*Mohr et al., 2018*; *Jonauskaite et al., 2020a*). In the current experiment, participants accessed the online survey on our laboratory computer to ensure comparability between the two experimental conditions.

The survey started with an information page. On the next pages, the task was explained, namely to associate colour terms with emotion concepts, presented on the GEW (see *Emotion assessment*). Participants had to perform a manipulation check exercise to make sure they understood the task. In particular, participants had to correct the responses of an imaginary person (Peter). In the following experimental part, participants saw the 12 colour terms written in black ink on a grey background, presented sequentially and in random order above the GEW (see *Colour stimuli* and Table 2). Participants were asked to

choose one, several, or none of the GEW emotion concepts that they associated with each colour term. They also rated intensities of each associated emotion by choosing circles of different sizes, which were later coded as 1-5 ratings. After the colour-emotion association task, participants provided demographic information and saw results from a previous related marketing experiment in graphic format.

*Colour patches condition*

We performed the colour patches condition in the same laboratory testing room as the terms condition. The experiment started with an information page explaining the task, namely to associate colour patches with emotion concepts, presented on the GEW (see *Emotion assessment*). Participants proceeded to the next page if they understood the task. Then, three example colours followed. For the examples as well as for the main task, participants were presented with a colour patch (15° × 15° subtended angle) on a neutral grey background (see Table 2). They were instructed to focus on the colour patch. Participants chose when to move to the subsequent page but no earlier than 5 s after it appeared on the screen. On each subsequent page, in analogy to the terms condition, participants associated one, several, or none of the GEW concepts with the target colour patch and rated the intensity of each associated emotion concept. While associating emotions, participants could see the target colour on the small GEW squares as well as on the chosen intensity circles (Fig. 2B Experiment 2). There were 12 experimental colour patches presented in randomised order (see *Colour stimuli* and Table 2). Colour values were adapted for the monitor (see *Apparatus* in Supplementary Material). We collected these data in the laboratory to ensure accurate colour presentation.

After the colour-emotion association task, participants completed the colour-naming task with the same colour patches. Each colour patch was presented 12 times in randomised order and paired with one of the colour terms (total of 144 presentations). Participants had to evaluate how likely they would be using this *colour term* to name a particular *colour patch* from ''not at all'' (converted to 0) to ''very likely'' (converted to 100). For example, participants would see a *green* colour patch and have to respond how likely they would be to call it *purple*. Not all participants in the patches condition performed the colour-naming task (22 colour-blind and 33 non-colour-blind completed the task). We decided to add this task after the first 10 colour-blind participants had been tested. After these two tasks, participants provided demographic information, analogous to the terms condition, on a paper questionnaire.

## Data preparation

The raw data can be accessed following this link: https://forsbase.unil.ch/project/study-public-overview/16969/0/. We cleaned the data based on colour blindness scores by creating the Colour Blindness Index.

*Colour Blindness Index*

We used errors on the colour blindness tests to create a single measure of colour blindness –the Colour Blindness Index. This index served a dual purpose. First, we could ensure
accurate participant re-categorisation into colour-blind and non-colour-blind participants. Second, we obtained a continuous measure of colour blindness.

To determine the colour blindness indices, we used a principal component analysis on the correlation matrix of the number of errors on the Ishihara test, the number of crossing errors on both the Farnsworth D-15 and Lanthony D-15 tests, and the number of neighbour errors on both the Farnsworth D-15 and Lanthony D-15 tests (see Supplementary Material for scoring). The principal component analysis resulted in two factors with Eigenvalues greater than 1 (i.e., 2.79 and 1.07 respectively for factors 1 and 2). The first factor explained 55.7% of the variance and the second factor explained an additional 21.3% of the variance. The first factor separated the colour-blind participants from the non-colour-blind participants, and we called this factor the Colour Blindness Index (see Fig. S1A). The second factor was difficult to interpret and did not separate participants by colour blindness (see Fig. S1B). Thus, we disregarded it. In Table S3, we present the loadings of each item for both factors.

The visual inspection of the frequency distribution of the Colour Blindness Index (Fig. S1A) indicates that it might consist of three different distributions. The most leftward distribution ($<-0.6$) included only non-colour-blind participants plus one colour-blind participant by self-report. Thus, the latter participant was most likely not colour-blind; he passed both the Farnsworth D-15 and Lanthony D-15 tests, and was categorised as "unsure" on the Ishihara test. The most rightward distribution ($>0.2$) included only colour-blind participants by self-report, thus, these participants had relatively strong colour blindness. The intermediate distribution (between $-0.6$ and $0.2$) included both self-reported colour-blind and non-colour-blind participants. Participants with these scores might have (very) weak colour blindness or no colour vision impairment but nevertheless made errors for other reasons (e.g., inattentiveness).

For the group-level analyses (see below), we considered only the two extreme groups (i.e., re-categorised non-colour-blind and re-categorised colour-blind participants). Such a categorisation ensured that participants grouped in the non-colour-blind group were indeed not colour-blind (had low Colour Blindness Index scores) while participants grouped in the colour-blind group were indeed relatively strongly colour-blind (i.e., had high Colour Blindness Index scores)[1]. There were 25 colour-blind and 25 non-colour-blind participants in the terms condition. There were 24 colour-blind and 31 non-colour-blind participants in the patches condition (see Table S4).

## Data analyses

We ran the subsequent analyses using these new and improved colour blindness categories. We set alpha levels for all tests at .050. All analyses were two-tailed. Across statistical tests, where appropriate, we controlled for familywise errors (Type I error) using False Discovery Rate (FDR) correction and marked the corrected $p$- values as $p_{FDR}$ (*Benjamini & Hochberg, 1995*). We performed analyses and created graphs with the R v.3.4.0 and SPSS v.25.

### Group-level analyses

For these analyses, we compared the re-categorised non-colour-blind and colour-blind participants, as described in the section Colour Blindness Index. We continue labelling them colour-blind and non-colour-blind participants, for simplicity.

[1]We chose the most inclusive limits. This allowed us to keep as many "real" non-colour-blind and "real" colour-blind participants as possible. However, less inclusive boundaries (i.e., excluding everyone who scored between $-0.7$ and $0.4$ on the Colour Blindness Index) did not change the overall results of our analyses and the respective conclusions. Please find the complete dataset at https://forsbase.unil.ch/project/study-public-overview/16969/0/.
*Specific colour-emotion associations.* We started the analyses by investigating the specific emotion concepts associated with colours. We calculated the proportion of participants who associated a specific emotion concept with a specific colour by dividing the number of participants who chose each emotion concept for each colour by the total number of participants in that group (e.g., colour-blind, terms condition). The proportion of participants was calculated separately for colour-blind participants and non-colour-blind participants for each condition (terms or patches) separately. The proportion values were the dependent variable, which varied from 0 (very unlikely association, no one chose it) to 1 (very likely association, everyone chose it).

To compare the *pattern* of emotion associations, we created four $12 \times 20$ (colours $\times$ emotions) representation matrices using the proportion values to compare colour blindness groups and colour presentation modes. $Matrix_{CB-term}$ contained colour-emotion associations of colour-blind participants associating colour terms with emotion concepts, while $Matrix_{Non-CB-term}$ contained analogous associations of non-colour-blind participants (terms condition). $Matrix_{CB-patch}$ contained colour-emotion associations of colour-blind participants associating colour patches with emotion concepts while $Matrix_{Non-CB-patch}$ contained analogous associations of non-colour-blind participants (patches condition; see Fig. 3).

Then, we used Pearson matrix correlations to compare $Matrix_{CB-term}$ vs. $Matrix_{Non-CB-term}$ and $Matrix_{CB-patch}$ vs. $Matrix_{Non-CB-patch}$. These matrix correlations formed the basis for the Pattern Similarity Index (PSI), which reflects the degree of similarity in the pattern of colour-emotion associations between two matrices. A PSI score of 1 indicates perfect pattern similarity, and a PSI score of 0 indicates complete pattern dissimilarity. Furthermore, to compare the similarity of emotion associations for each colour, we calculated $PSI_{colour}$. $PSI_{colour}$ was estimated per colour using Pearson correlations between colour-blind participants and non-colour-blind participants, and between colour terms and colour patches.

To identify which colour-emotion associations differed between colour-blind and non-colour-blind participants, we further used Fisher's exact tests (*Fisher, 1922*). The test compared the proportion of participants endorsing a particular colour-emotion association (yes/no; $n = 240$) between colour-blind and non-colour-blind participants for terms and for patches separately. All comparisons were FDR corrected (*Benjamini & Hochberg, 1995*).

*Emotion intensity.* The dependent variable *emotion intensity* was calculated by averaging intensity ratings assigned to emotion concepts associated with each colour and for any colour (i.e., "overall"). Emotion intensity varied from 1 (weak) to 5 (strong), unless no emotion concept was chosen (coded as missing value).

A $2 \times 2$ independent-samples ANOVA compared average *emotion intensity* of all colours together (i.e., "overall") between re-categorised study groups (colour-blind vs. non-colour-blind) and conditions (colour terms vs. colour patches). Afterwards, series of independent-samples *t*-tests compared *emotion intensity* ratings per colour between colour-blind and non-colour-blind participants for terms and for patches separately, and between terms and

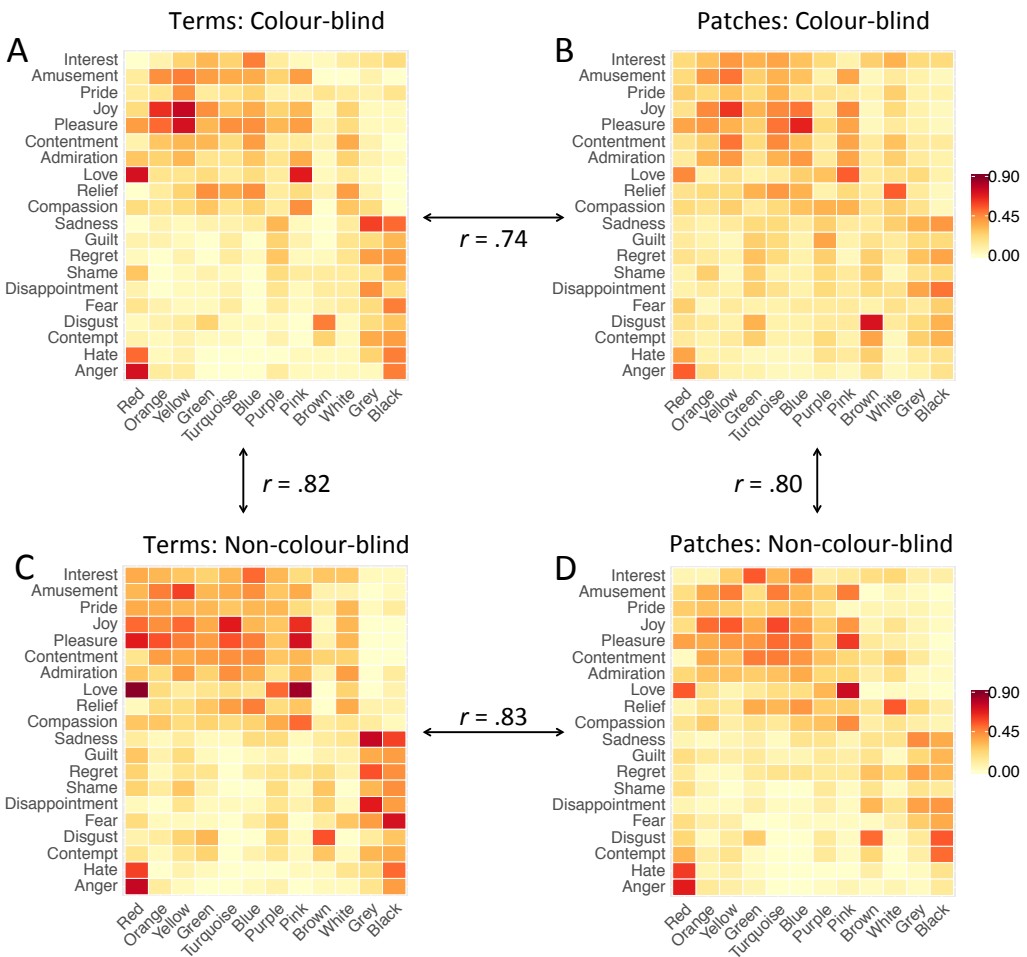

**Figure 3 Heatmaps of colour-emotion associations.** (A) Colour-emotion associations with colour terms in colour-blind participants; (B) Colour-emotion associations with colour patches in colour-blind participants; (C) colour-emotion associations with colour terms in non-colour-blind participants; (C) colour-emotion associations with colour patches in non-colour-blind participants. Redder cells indicate higher proportions of participants choosing these specific colour-emotion associations.

patches for colour-blind and non-colour-blind participants separately. All comparisons were FDR corrected (*Benjamini & Hochberg, 1995*).

*Supplemental analyses* Additionally, we analysed colour associations with emotion dimensions and colour naming. As these were supplementary analyses, the method and results are presented in the Supplementary Material.

### Individual-level analyses

We tested whether the *presence* of colour-emotion associations depended on the degree of colour blindness (i.e., Colour Blindness Index). To this end, we fitted a logistic model with repeated measures using *glmer* function in lme4 package (*Bates et al., 2015*). We used the likelihood ratio test to test for significance of individual predictors and their interactions.

The predictor variables were the Colour Blindness Index, colour presentation mode (2 levels, independent), colour (12 levels, repeated), and emotion (20 levels, repeated). We also tested for the three interactions between the Colour Blindness Index and (i) colour presentation mode, (ii) colour, and (iii) emotion. The outcome variable was presence of colour-emotion association (yes/no). These analyses were performed on all participants ($n = 129$).

## RESULTS

### Colour blindness scoring

The $2 \times 2$ MANOVA on the number of errors in colour blindness test scores indicated that colour-blind participants made significantly more errors than non-colour-blind participants on all tests. However, their performance did not differ by condition (terms or patches). More details are presented in the Supplementary Material and Table S5.

### Group-level analyses
#### *Specific colour-emotion associations*

Some colour-emotion associations were prominent in both study groups. For instance, *red-love*, *red-anger*, *yellow-joy*, *pink-love* and *brown-disgust* were chosen by 50% or more of colour-blind as well as non-colour-blind participants (terms and patches combined). The majority of colour-blind participants also associated *orange* with *joy*, *yellow* with *pleasure*, and *blue* with *pleasure*. The majority of non-colour-blind participants also associated *red* with *pleasure*, *red* with *hate*, *yellow* with *amusement*, *turquoise* with *joy* and *pleasure*, *blue* with *interest*, *pink* with *joy* and *pleasure*, *grey* with *sadness* and *disappointment,* and *black* with *fear*. See Fig. 3 for visual representation of all colour-emotion associations (and supplemental tables for the numeric values: Table S10, Table S11, Table S12, and Table S13).

*Colour-blind vs. non-colour-blind participants* After having described the specific colour-emotion associations, we compared the pattern of colour-emotion associations between study groups and conditions. The matrix correlations, PSI, were overall high. PSI comparing emotion associations with colour terms ($Matrix_{CB-term}$ vs. $Matrix_{Non-CB-term}$) showed high similarity, $r = .82$, $R^2 = .672$, $p < .001$, and so did PSI comparing emotion associations with patches ($Matrix_{CB-patch}$ vs. $Matrix_{Non-CB-patch}$), $r = .80$, $R^2 = .637$, $p < .001$ (see Fig. 3). These correlation coefficients were of similar strength, $z = -0.63$, $p = .529$. These results imply that colour-blind participants and non-colour-blind participants associated similar emotions, irrespective of whether a colour was presented as a term or a patch.

Furthermore, colour-specific $PSI_{colour}$ comparing emotion associations between colour-blind and non-colour-blind participants for each colour were high for colour terms, $r = .60$-.97, $R^2 = .355 - .939$, $p_{FDR} < .006$, and for colour patches, $r = .55$-.92, $R^2 = .548 - .924$, $p_{FDR} < .012$, see Table 3. The only exception was *purple*, for colour terms, $r = .07$, $R^2 = .004$, $p_{FDR} = .781$, and for colour patches, $r = .09$, $R^2 = .007$, $p_{FDR} = .721$. These results indicate that the similarity between colour-blind and non-colour-blind participants held across all colours, whether a term or a patch was presented, with *purple* being an exception.

**Table 3  Matrix-to-matrix correlations per colour (PSI$_{colour}$), separated by correlations between colour-blind and non-colour-blind participant association matrices, and between colour terms and colour patches association matrices.**

| | Colour blind vs. Non-colour-blind | | Terms vs. Patches | |
| --- | --- | --- | --- | --- |
| | **Terms** | **Patches** | **Colour-blind** | **Non-colour-blind** |
| Red | 0.88*** | 0.85*** | 0.84*** | 0.82*** |
| Orange | 0.85*** | 0.77*** | 0.83*** | 0.85*** |
| Yellow | 0.84*** | 0.90*** | 0.83*** | 0.88*** |
| Green | 0.80*** | 0.55* | 0.46* | 0.76*** |
| Turquoise | 0.83*** | 0.92*** | 0.87*** | 0.95*** |
| Blue | 0.97*** | 0.86*** | 0.84*** | 0.96*** |
| Purple | 0.07 | 0.09 | 0.26 | 0.69** |
| Pink | 0.90*** | 0.87*** | 0.89*** | 0.95*** |
| Brown | 0.79*** | 0.82*** | 0.82*** | 0.84*** |
| Grey | 0.91*** | 0.76*** | 0.86*** | 0.89*** |
| White | 0.60** | 0.88*** | 0.75*** | 0.49* |
| Black | 0.92*** | 0.86*** | 0.67** | 0.68** |

**Notes.**

The PSI$_{colour}$ (correlation coefficient $r$) indicates the similarity between two matrices with 1 indicating perfect similarity. All $p$-values are FDR corrected for multiple comparisons.

*$p < .050$.
**$p < .010$.
***$p < .001$.

Fisher's exact tests were used to identify any differences between the specific colour-emotion associations between the two study groups, separately for each condition. No specific colour-emotion comparisons were significant suggesting that no specific colour-emotion association differed between the two study groups ($p_{FDR} \geq .39$). Thus, despite low correlations for *purple*, we could not detect specific emotion associations driving this dissimilarity.

*Colour terms vs. colour patches.*  Furthermore, we compared the patterns of emotion associations with colour terms and colour patches, respectively, for each study group separately. The matrix-to-matrix correlations, PSI, were again overall high. PSI comparing emotion associations between colour terms and colour patches in colour-blind participants (Matrix$_{CB-term}$ vs. Matrix$_{CB-patch}$) showed high similarity, $r = .74$, $R^2 = .552$, $p < .001$, and so did PSI comparing emotion associations between colour terms and colour patches in non-colour-blind participants (Matrix$_{Non-CB-term}$ vs. Matrix$_{Non-CB-patch}$), $r = .83$, $R^2 = .683$, $p < .001$ (see Fig. 3). However, the correlation coefficient in colour-blind participants was significantly lower than in non-colour-blind participants, $z = -2.59$, $p = .010$. These results mean that similar emotions were associated with colour terms and with colour patches by non-colour-blind participants as well as by colour-blind participants, but the latter did so to a lower extent.

Furthermore, colour-specific PSI$_{colour}$ comparing emotion associations between colour terms and colour patches for each colour were high for colour-blind participants, $r = .46 - .89$, $R^2 = .214 - .795$, $p_{FDR} < .040$, and for non-colour-blind participants, $r = .49 - .96$, $R^2 = .243 - .929$, $p_{FDR} < .027$, see Table 3. The exception again was *purple*,

associations of which did not correlate for colour-blind participants, $r = .26$, $R^2 = .066$, $p_{FDR} = .273$. Correlations for *green* in colour-blind participants were significant but low ($p = .040$). These results indicated that the similarity between colour terms and colour patches was equally true for colour-blind and non-colour-blind participants, with the exception of *purple*.

Fisher's exact tests were used to identify differences for specific colour-emotion associations between conditions, separately for colour-blind and non-colour-blind participants. No specific colour-emotion comparisons were significant ($p_{FDR} \geq .57$). Thus, despite a low correlation in colour-blind participants between *purple* as a patch and as a term, we could not detect specific emotion associations driving this dissimilarity.

### *Emotion intensity*

The $2 \times 2$ ANOVA revealed a significant main effect of condition, $F (1, 101) = 14.8$, $p < .001$, $\eta_p^2 = .123$, indicating that more intense emotions were associated with colour terms than colour patches by both study groups. There was no significant main effect of study group, $F (1, 101) = 2.44$, $p = .121$, $\eta_p^2 = .024$, indicating that colour blind and non-colour-blind participants associated equally intense emotions overall. Finally, the interaction between study group and condition was not significant, $F (1, 101) = 0.23$, $p = .440$, $\eta_p^2 = .006$. For differences by colour, see Supplemental Material (Table S6, Table S7, Table S8, and Table S9).

### Individual-level analyses

The multilevel logistic regression model was overall significant, $LR (63) = 876$, $p < .001$, $_{pseudo}R^2 = .028$ (Cox & Snell), $.047$ (Nagelkerke). Both, colour, $LR (12) = 161$, $p < .001$, $_{pseudo}R^2 = .005$ (Cox & Snell), $.009$ (Nagelkerke), and emotion, $LR (20) = 675$, $p < .001$, $_{pseudo}R^2 = .022$ (Cox & Snell), $.037$ (Nagelkerke), were significant predictors of whether colours and emotions were associated or not. In contrast, the Colour Blindness Index was not a significant predictor of the probability of colour-emotion associations, $LR (1) = 0.03$, $p = .865$, $_{pseudo}R^2 < .001$ (Cox & Snell), $< .001$ (Nagelkerke). Hence, the probability of colour-emotion associations did not vary by degree of colour blindness. Condition was not a significant predictor either, $LR (1) = 0.14$, $p = .711$, $_{pseudo}R^2 < .001$ (Cox & Snell), $< .001$ (Nagelkerke).

The two-way interaction between the Colour Blindness Index and colour was significant, $LR (11) = 23.4$, $p = .016$, $_{pseudo}R^2 = ..001$ (Cox & Snell), $.001$ (Nagelkerke). Higher Colour Blindness Index resulted in lower probability of emotion associations with *red*, $\beta = -0.17$, $z = -2.08$, $p = .037$. However, this effect was weak and disappeared after FDR correction ($p_{FDR} = .44$). The Colour Blindness Index was not a significant predictor for other colours, $ps_{FDR} = .96$. The other two-way interactions between the Colour Blindness Index and emotion, $LR (19) = 9.58$, $p = .96$, $_{pseudo}R^2 < .001$ (Cox & Snell), $<.001$ (Nagelkerke), and the Colour Blindness Index and condition, $LR (1) = 1.73$, $p = .189$, $_{pseudo}R^2 < .001$ (Cox & Snell), $<.001$ (Nagelkerke), were not significant.

Given these zero results, we wished to estimate the likelihood that, indeed, the Colour Blindness Index is unlikely to predict the probability of colour-emotion associations. We

examined the key predictor of interest (Colour Blindness Index) by estimating the Bayes factor using Bayesian Information Criteria (*Wagenmakers, 2007*; *Jarosz & Wiley, 2014*). The Bayes factor compared the fit of the data under the null hypothesis with the fit of the data under the alternative hypothesis. The estimated Bayes factor (null/alternative; $BF_{01}$) was 245:1, suggesting that the data were 245 times more likely to occur under the null hypothesis than the alternative hypothesis. Reversely, the data were 0.004 times more likely to occur under the alternative than the null hypothesis ($BF_{10}$).

## DISCUSSION

Colours are associated with emotions (*Wexner, 1954*; *Adams & Osgood, 1973*; *Valdez & Mehrabian, 1994*; *Kaya & Epps, 2004*; *Fugate & Franco, 2019*; *Tham et al., 2019*; *Schloss, Witzel & Lai, 2020*) and these associations might be universal across cultures (*Adams & Osgood, 1973*; *D'Andrade & Egan, 1974*; *Gao et al., 2007*; *Ou et al., 2018*; *Jonauskaite et al., 2020a*). If the assumption on universality holds true, we have to ask whether these associations originate from our shared (i) conceptual, abstract understanding of the world (*Xu, Dowman & Griffiths, 2013*), or (ii) perceptual experience of inhabiting the globe (*Palmer & Schloss, 2010*). Recently, *Jonauskaite et al. (2020b)* showed that colour-emotion associations were similar for colour patches and colour terms in young Swiss adults. These results indicate that (i) conceptual colour experiences seem sufficient for colour-emotion associations to be reported, and (ii) immediate perceptual colour experiences do not seem necessary.

To further assess these suggestions, we tested men with congenital red-green colour blindness as well as men with intact colour vision. We tested men, because they have a much higher incidence of colour blindness than women (*Sharpe et al., 1999*; *Birch, 2012*). Our participants associated 12 colours with 20 emotion terms, and rated emotion intensities (see also *Jonauskaite et al., 2020b*). Half of our participants associated colour terms, and the other half associated colour patches. Participants who associated colour patches also named them. We found that colour-blind and non-colour-blind men showed a high degree of similarity in colour-emotion associations, whether associating colour terms or colour patches. In case of colour patches, the two groups named colours almost identically. Furthermore, the strength of colour blindness neither predicted colour-emotion associations nor emotion intensities. Within group comparisons showed highly similar emotion associations with terms and patches (see also *Jonauskaite et al., 2020b*), with yet a higher similarity found in non-colour-blind than colour-blind men.

Before discussing these major findings, we highlight that we tested representative samples. We replicated common colour-emotion associations such as *red-love, red-anger, yellow-joy, pink-love,* and *brown-disgust* associations (*Kaya & Epps, 2004*; *Fugate & Franco, 2019*; *Jonauskaite et al., 2019a*; *Jonauskaite et al., 2020a*). When we clustered the 20 emotion concepts into the affective dimensions of valence, arousal, and power, we replicated that *black*, *grey*, and *brown* were negative colours; *yellow, orange, blue, turquoise, pink,* and *white* were positive colours; and *red* was an arousing and powerful colour associated with both positive and negative emotions (*Adams & Osgood, 1973*; *Valdez & Mehrabian,*

*1994*; *Soriano & Valenzuela, 2009*; *Lakens, Semin & Foroni, 2012*; *Sutton & Altarriba, 2016*; *Specker et al., 2018*; *Jonauskaite et al., 2020b*). These colour-emotion associations were endorsed by both colour-blind and non-colour-blind men.

When returning to our major findings, we have to first remember that colour-blind individuals perceive colours differently from non-colour-blind individuals since birth (*Linhares, Pinto & Nascimento, 2008*). They have diminished or completely absent excitations of the L or M photoreceptors (*Dalton, 1798*; *Parry, 2015*). Second, we have to remember that colour-blind individuals have learned the same conceptual representations of colour as non-colour-blind individuals (*Byrne & Hilbert, 2010*), including colour naming (*Bonnardel, 2006*, and the current study). With these pieces of information in mind, we can start considering what it might mean that our colour-blind and non-colour-blind participants provided highly similar colour-emotion associations, despite partially different perceptual experiences. First of all, participants likely activated similar abstract colour representations when reading a colour term (e.g., *red*) to when looking at the actual colour patch. Then, we can also consider that the colour-emotion associations were more majorly driven by the conceptual representations of colours, because seeing actual colour patches seemed to carry no additional information to colour-emotion associations (see also, *Jonauskaite et al., 2020b*). The latter consideration echoes analogue notions for colour-tone associations (*Saysani, 2019*), transmission of colour terms (*Xu, Dowman & Griffiths, 2013*), mental colour spaces (*Shepard & Cooper, 1992*; *Saysani, Corballis & Corballis, 2018a*; *Saysani, Corballis & Corballis, 2018b*), or object-colour knowledge (*Wang et al., 2020*). So far, we have to limit our reasoning to colour-emotion associations for focal colours, which we presented here, and which are highly recognisable by colour-blind men (see also, *Moreira et al., 2014*).

So far, we have discussed the high similarities between groups and conditions. However, the degree of similarities fell short of 100%, leaving space for additional variance to be explained. Part of this variance might be random noise, but part might be linked to meaningful individual differences. In this regard, the degree of colour blindness was uninformative; it did not explain colour-emotion associations or emotion intensities. We observed, however, that the similarity of emotion associations with terms and patches was less pronounced for colour-blind than non-colour-blind men. This relatively lower similarity points to a possible influence of actual colour experiences to colour-emotion associations (see also, *Saysani, Corballis & Corballis, 2018b*; *Shepard & Cooper, 1992*). One could suggest that colour-blind men as compared to non-colour-blind men were less certain when naming colour patches. This suggestion seems unlikely, however, because colour-blind and non-colour-blind men named the patches of focal colours almost identically. Alternatively, due to perceptual deficiencies, colour-blind men who saw colour patches might have activated slightly different abstract colour representations than colour-blind men who read colour terms, especially for colours affected by colour blindness. We found that colour-blind men showed the lowest patch-term similarities for *purple* and *green,* and associated more intense emotion concepts with *red*, *orange*, *yellow*, *pink*, *black*, and *white* when colours were presented as terms than patches (see also *Jonauskaite et al., 2020b*) for stronger emotion intensities with terms than patches). Also, colour-blind men associated

fewer emotion concepts with *red* than non-colour-blind men. Colour-blind men might have imagined these colours more vividly than seen in patches, associating more intense and specific emotions when processing these terms.

Overall, our observations on high degrees of similarities support the previous literature, showing high similarities in colour-emotion associations across cultures (*Adams & Osgood, 1973*; *D'Andrade & Egan, 1974*; *Gao et al., 2007*; *Ou et al., 2018*; *Jonauskaite et al., 2019c*; *Jonauskaite et al., 2020a*). At the same time, studies have also shown systematic variations on long-term and short-term scales. On long-term scales, high similarities in colour-emotion associations were more pronounced when individuals came from nations that were linguistically and/or geographically closer (*Jonauskaite et al., 2020a*). For instance, individuals living closer to the equator had a lower likelihood to associate *yellow* with *joy* than individuals living further away from the equator (*Jonauskaite et al., 2019a*). Studies have also shown systematic variations on shorter time scales. Individuals living in the same nation preferred autumn-like colours more strongly in autumn than during other seasons of the year (*Schloss & Heck, 2017*; *Schloss et al., 2017*). On even shorter time scales, colour preferences have been influenced in a laboratory experiment (*Strauss, Schloss & Palmer, 2013*). These authors showed that exposure to numerous positive objects (e.g., strawberries and wine) increased the liking of the respective colour (e.g., *red*), while exposure to numerous negative objects (e.g., a bloody nose and rotten tomatoes) decreased the liking of the respective colour (e.g., *red*). Likely, studies showing such systematic variations demonstrate the human species' abilities to adapt to particularities of their respective environments (*Lupyan & Dale, 2016*).

As an auxiliary finding, we observed a low similarity in emotion associations with *purple*. We observed dissimilar associations between colour-blind and non-colour-blind men as well as between terms and patches in colour-blind men. Colour-blind men associated *purple* with diverse positive as well as negative emotions, while non-colour-blind men associated *purple*, especially as a term, with positive emotions, mainly with *love*. Diverse findings for *purple* are not new. Participants in general disagree which emotions *purple* represent, whether data originate from the same nation (*Wexner, 1954*; *Hemphill, 1996*; *Sandford, 2014*; *Sutton & Altarriba, 2016*; *Fugate & Franco, 2019*), from four or 30 nations (*Hupka et al., 1997*; *Jonauskaite et al., 2020a*), or when comparing terms and patches (*Jonauskaite et al., 2020b*), as was also done here. We suggest that this lack of clarity for *purple* is an interesting observation, so much so that it deserves its own investigation (e.g., *Hamilton, 2014*; *Oja & Uusküla, 2011*; *Tager, 2018*).

## Strengths and limitations

There are numerous strengths and limitations to our study. The first strength is that we employed the same method used previously to assess colour-emotion associations (*Griber, Jonauskaite & Mohr, 2019*; *Jonauskaite et al., 2019a*; *Jonauskaite et al., 2019b*; *Jonauskaite et al., 2019c*; *Jonauskaite et al., 2020a*; *Jonauskaite et al., 2020b*). This consistency simplifies direct comparisons between studies. The second strength is that we recruited a large number of congenitally colour-blind men, at least when comparing our sample size to previous studies (*Shepard & Cooper, 1992*; *Paramei, 1996*; *Paramei, Bimler & Cavonius,*

*1998*; *Bonnardel, 2006*; *Moreira et al., 2014*; *Álvaro et al., 2015*; *Álvaro et al., 2017*; *Sato & Inoue, 2016*; *Saysani, Corballis & Corballis, 2018a*). By default, a larger sample size provides more representative colour-emotion associations. Yet, having a larger sample size for our colour-blind men also meant that our sample was relatively diverse (see also *Bonnardel, 2006*; *Nagy & Ábrahám, 2014*; *Paramei, 1996*). We recruited all men who had self-reported congenital red-green colour blindness, irrespective of its strength. Thus, we tested men with partial as well as complete colour vision deficiencies (i.e., dichromatic and anomalous trichromatic vision), with mainly deutan-like or unidentified impairments. Only some previous studies aimed for a sample of exclusively dichromatic participants (e.g., *Álvaro et al., 2015*; *Álvaro et al., 2017*; *Moreira et al., 2014*; *Shepard & Cooper, 1992*), resulting in a much smaller number of tested individuals.

To factor in this diversity and to account for varying strength of colour blindness, we derived the Colour Blindness Index from scores on three behavioural colour vision tests (*Farnsworth, 1947*; *Lanthony, 1978b*; *Ishihara, 2000*). This Colour Blindness Index was not a significant predictor of colour-emotion associations, while between- as well as within-group similarities were high. Therefore, we argue that differences in colour perception within our colour-blind group bore little relevance to colour-emotion associations, at least when working with highly recognisable focal colours. If this conclusion holds true, similar colour-emotion associations should also arise in congenitally blind individuals. Previous studies have demonstrated that congenitally blind individuals possess similar mental spaces of colour (*Saysani, Corballis & Corballis, 2018b*), associate similar colours with pure tones (*Saysani, 2019*), and represent object-colour knowledge in similar brain regions as sighted individuals (*Wang et al., 2020*). Some blind individuals also associate similar colours with semantic scales, but there is a high variability among the blind (*Saysani, Corballis & Corballis, 2021*).

Another potential limitation is the use of focal colours (i.e., best examples of colour categories) and basic colour terms, both of which are overlearned. Testing colour patches that are difficult to name or using non-basic colour terms, like *lavender* or *mauve*, would be the next step in this type of research. Such colour stimuli might be more powerful to reveal more differences between colour-blind and non-colour-blind individuals. The perceptual experience might be more important when working with stimuli that are less overlearned. In a previous study (*Saysani, Corballis & Corballis, 2018a*), the mental arrangement of non-basic colour terms was less similar between colour-blind and non-colour-blind individuals than the mental arrangement of the basic colour terms. Yet, the similarity between the two groups was still very high in both conditions, suggesting that colour-blind participants have a common understanding of non-basic colour terms too.

## Theoretical and practical implications

All results considered, we conclude that cultural knowledge, transmitted through language, plays a sufficient role for colour-emotion associations to be reported, while immediate perceptual colour experience in adulthood does not seem to be necessary. This conclusion has implications to theories in which the importance of colour perception to affective associations with colour is highlighted (*Hurlbert & Ling, 2007*; *Palmer & Schloss,*

*2010*; *Schloss, 2018*). According to the cone-opponent theory (*Hurlbert & Ling, 2007*), human colour preferences are influenced by weights on the two cone-opponent contrast components (i.e., L-M; S-(L+M)). According to the Ecological Valence Theory (*Palmer & Schloss, 2010*), human colour preferences are driven by the valence of objects of the same colour. For instance, people like colours that are associated with positive objects and dislike colours that are associated with negative objects. As an example, *blue* would be liked because it is associated with clear sky and clean water while *brown* would be disliked because it is associated with rotten food. Note, these theories have been developed to explain colour preferences and not colour-emotion associations (but see *Schloss, 2018*). Perhaps, colour preferences and colour-emotion associations are guided by different mechanisms. In fact, colour preferences have been hypothesised (*Schloss, 2015*) and empirically demonstrated (*Álvaro et al., 2015*; *Baek et al., 2015*; *Sato & Inoue, 2016*) to differ between colour-blind and non-colour-blind individuals. More specifically, colour-blind individuals preferred *yellowish* colours to a greater extent and *bluish* colours to a lesser extent than non-colour-blind individuals (*Álvaro et al., 2015*). Colour preferences seem also less universal (*Taylor, Clifford & Franklin, 2013*; *Schloss & Palmer, 2017*; *Groyecka et al., 2019*). Thus, immediate perceptual experiences might be more relevant to colour preferences than to colour-emotion associations. Alternatively, future theories should account for more conceptual, knowledge- and language-based factors when explaining colour preferences (see *Yokosawa et al., 2016* for the importance of symbolic colour associations to colour preferences).

If immediate perceptual experiences are not necessary for colour-emotion associations in adulthood, then research on colour-emotion associations might not easily translate to applied domains. For instance, proponents of colour therapy, or chromotherapy, assume that perception of colour can impact one's affective states (*Azeemi & Raza, 2005*; *O'Connor, 2011*; *Winkler, 2012*; *Gul, Nadeem & Aslam, 2015*). Often, such claims are based on conceptual colour associations. One can read, "Being the lightest hue of the spectrum, the colour psychology of yellow is uplifting and illuminating, offering hope, happiness, cheerfulness and fun" (*Scott-Kemmis, 2018*). *Yellow* was indeed conceptually associated with *joy* in 55 countries (*Jonauskaite et al., 2019a*). However, an association between *yellow* and *joy* does not immediately imply that looking at *yellow* walls or *yellow* objects would make one feel *joyful*. Empirical studies have struggled to confirm many of the expected psychological effects of colour, such as *pink* reducing aggressiveness in prisoners (*Genschow et al., 2015*), or *pink*, *red*, or *blue* enhancing cognitive performance and improving mood (*Von Castell et al., 2018*). A recent study also demonstrated that direct exposure to colour was not important to stress and anxiety reduction following a colour intervention (*Jonauskaite et al., 2020c*). In short, conceptual colour-emotion associations should not be equated with and might not translate to psychological consequences of colour.

## CONCLUSIONS

We evaluated whether conceptual mechanisms are sufficient for consistent colour-emotion associations to be reported or whether immediate colour experience is necessary. We found

that colour-emotion associations were highly similar between individuals with congenital red-green colour blindness and individuals with intact colour vision. This high similarity was observed whether colours were shown as terms or patches. Based on our findings, we conjecture that intact immediate colour vision is not necessary for colour-emotion associations to be reported, at least not in adulthood. Likely, these associations are driven by conceptual mechanisms, our language and knowledge. In other words, it is unlikely that colour-emotion associations arise exclusively from direct affective experiences when seeing colours, because conceptual knowledge is already well established. To reason one step further, high similarities between colour-blind and non-colour-blind individuals as well as similarities across cultures (*Adams & Osgood, 1973*; *Jonauskaite et al., 2020a*) would suggest that colour-emotion associations present another human psychological universal (*Norenzayan & Heine, 2005*).

## ACKNOWLEDGEMENTS

We wish to thank Amer Chamseddine for programming the first experiment and Guillaume Sierro for programming the second experiment.

### Funding

This research was supported with a Doc.CH fellowship grant to Domicele Jonauskaite (P0LAP1_175055) and project grant to Christine Mohr (100014_182138) from the Swiss National Science Foundation. C. Alejandro Parraga was funded by the 2017-SGR-649 and DPI2017-89867-C2-1-R grants from the Catalan and the Spanish Science Ministries. The funders had no role in study design, data collection and analysis, decision to publish, or preparation of the manuscript.

### Grant Disclosures

The following grant information was disclosed by the authors:
Doc.CH fellowship grant to Domicele Jonauskaite:  P0LAP1_175055.
Swiss National Science Foundation:  100014_182138).
Catalan and the Spanish Science Ministries: 2017-SGR-649,  DPI2017-89867-C2-1-R.

### Competing Interests

The authors declare there are no competing interests.

### Author Contributions

- Domicele Jonauskaite conceived and designed the experiments, performed the experiments, analyzed the data, prepared figures and/or tables, authored or reviewed drafts of the paper, and approved the final draft.
- Lucia Camenzind performed the experiments, analyzed the data, authored or reviewed drafts of the paper, and approved the final draft.
- C. Alejandro Parraga and Christine Mohr conceived and designed the experiments, authored or reviewed drafts of the paper, and approved the final draft.

- Cécile N. Diouf, Mathieu Mercapide Ducommun, Lauriane Müller and Mélanie Norberg performed the experiments, authored or reviewed drafts of the paper, and approved the final draft.

### Human Ethics

The following information was supplied relating to ethical approvals (i.e., approving body and any reference numbers):

The study was approved by the Research Ethics Commission of the University of Lausanne (Ethics Application Ref: C-SSP-032020-00003).

### Data Availability

The raw data are available on FORSbase:

Domicele Jonauskaite, Christine Mohr: Colour-emotion associations in individuals with and without congenital red-green colour-blindness [Dataset]. University of Lausanne - Faculty of Social and Political Sciences - Institute of Psychology - Cognitive and Affective Regulation Laboratory - CARLA. Distributed by FORS, Lausanne, 2021. https://doi.org/10.23662/FORS-DS-1230-2.

One must create a free account on FORSbase, after which they can download the data without restrictions.

### Supplemental Information

Supplemental information for this article can be found online at http://dx.doi.org/10.7717/peerj.11180#supplemental-information.

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

# PeerJ

**Norenzayan A, Heine SJ. 2005.** Psychological universals: what are they and how can we know?. *Psychological Bulletin* **131**:763–784 DOI 10.1037/0033-2909.131.5.763.

**O'Connor Z. 2011.** Colour psychology and colour therapy: Caveat emptor. *Color Research and Application* **36**:229–234 DOI 10.1002/col.20597.

**Oja V, Uusküla M. 2011.** Indo-European influence on Finnic colour naming and categorisation. In: Varella S, ed. *Languages and Cultures in contact and contrast: historical and contemporary perspectives.* Athens: Atiner, 7–18.

**Ou L-C, Yuan Y, Sato T, Lee W-YY, Szabó F, Sueeprasan S, Huertas R. 2018.** Universal models of colour emotion and colour harmony. *Color Research and Application* **43**:736–748 DOI 10.1002/col.22243.

**Palmer SE, Schloss KB. 2010.** An ecological valence theory of human color preference. *Proceedings of the National Academy of Sciences* **107**:8877–8882 DOI 10.1073/pnas.0906172107.

**Paramei GV. 1996.** Color space of normally sighted and color-deficient observers reconstructed from color naming. *Psychological Science* **7**:311–317 DOI 10.1111/j.1467-9280.1996.tb00380.x.

**Paramei GV, Bimler DL, Cavonius CR. 1998.** Effect of luminance on color perception of protanopes. *Vision Research* **38(97)**:3397–3401 DOI 10.1016/S0042-6989(97)00454-9.

**Parry NRA. 2015.** Color vision deficiencies. In: Elliot AJ, Fairchild MD, Franklin A, eds. *Handbook of color psychology.* Cambridge: Cambridge University Press, 216–242 DOI 10.1017/CBO9781107337930.011.

**Sandford JL. 2014.** Turn a colour with emotion: a linguistic construction of colour in English. *Journal of the International Colour Association* **13**:67–83.

**Sato K, Inoue T. 2016.** Perception of color emotions for single colors in red-green defective observers. *PeerJ* **4**:e2751 DOI 10.7717/peerj.2751.

**Saysani A. 2019.** How the blind hear colour. *Perception* **48**:237–241 DOI 10.1177/0301006619830940.

**Saysani A, Corballis MC, Corballis PM. 2018a.** The colour of words: how dichromats construct a colour space. *Visual Cognition* **26**:601–607 DOI 10.1080/13506285.2018.1524804.

**Saysani A, Corballis MC, Corballis PM. 2018b.** Colour envisioned: concepts of colour in the blind and sighted. *Visual Cognition* **26**:382–392 DOI 10.1080/13506285.2018.1465148.

**Saysani A, Corballis MC, Corballis PM. 2021.** Seeing colour through language: colour knowledge in the blind and sighted. *Visual Cognition* **29**:63–71 DOI 10.1080/13506285.2020.1866726.

**Scherer KR. 2005.** What are emotions? And how can they be measured?. *Social Science Information* **44**:695–729 DOI 10.1177/0539018405058216.

**Scherer KR, Shuman V, Fontaine JRJ, Soriano C. 2013.** The GRID meets the Wheel: assessing emotional feeling via self-report. In: Fontaine JRJ, Scherer KR, Soriano C, eds. *Components of emotional meaning: a sourcebook.* Oxford: Oxford University Press, 281–298 DOI 10.13140/RG.2.1.2694.6406.

**Schloss KB. 2015.** Color preferences differ with variations in color perception. *Trends in Cognitive Sciences* **19**:554–555 DOI 10.1016/j.tics.2015.08.009.

**Schloss KB. 2018.** Chapter 6. A color inference framework. *Progress in Colour Studies* 107–122 DOI 10.1075/z.217.06sch.

**Schloss KB, Heck IA. 2017.** Seasonal changes in color preferences are linked to variations in environmental colors: a longitudinal study of fall. *i-Perception* **8**:1–19 DOI 10.1177/2041669517742177.

**Schloss KB, Nelson R, Parker L, Heck IA, Palmer SE. 2017.** Seasonal variations in color preference. *Cognitive Science* **41**:1589–1612 DOI 10.1111/cogs.12429.

**Schloss KB, Palmer SE. 2017.** An ecological framework for temporal and individual differences in color preferences. *Vision Research* **141**:95–108 DOI 10.1016/j.visres.2017.01.010.

**Schloss KB, Witzel C, Lai LY. 2020.** Blue hues don't bring the blues: questioning conventional notions of color–emotion associations. *Journal of the Optical Society of America A* **37**:813–824 DOI 10.1364/JOSAA.383588.

**Scholtyssek C, Kelber A, Dehnhardt G. 2014.** Why do seals have cones? Behavioural evidence for colour-blindness in harbour seals. *Animal Cognition* **18**:551–560 DOI 10.1007/s10071-014-0823-3.

**Scott-Kemmis J. 2018.** The color yellow. *Available at https://www.empower-yourself-with-color-psychology.com/color-yellow.html* (accessed on 17 April 2020).

**Sharpe LT, Stockman A, Jägle H, Nathans J. 1999.** In: Gegenfurtner KR, Sharpe LT, eds. *Opsin genes, cone photopigments, color vision, and color blindness.* Cambridge: Cambridge University Press, 3–52.

**Shepard RN, Cooper LA. 1992.** Representation of colors in the blind, color-blind, and normally sighted. *Psychological Science* **3**:97–104 DOI 10.1111/j.1467-9280.1992.tb00006.x.

**Siniscalchi M, D'Ingeo S, Fornelli S, Quaranta A. 2017.** Are dogs red–green colour blind? *Royal Society Open Science* **4**:170869 DOI 10.1098/rsos.170869.

**Soriano C, Fontaine JRJ, Scherer KR, Akırmak GA, Alarcón P, Alonso-Arbiol I, Bellelli G, Pérez-Aranibar CC, Eid M, Ellsworth P, Galati D, Hareli S, Hess U, Ishii K, Jonker C, Lewandowska-Tomaszczyk B, Meiring D, Mortillaro M, Niiya Y, Ogarkova A, Panasenko N, Ping H, Protopapas A, Realo A, Ricci-Bitti PE, Shen Y-L, Sheu C-F, Siiroinen M, Sunar D, Tissari H, Tong EMW, Osch Yvan, Wong S, Yeung DY, Zitouni A. 2013.** Cross-cultural data collection with the GRID instrument. In: *Components of emotional meaning: a sourcebook.* Oxford: Oxford University Press, 98–105 DOI 10.1093/acprof:oso/9780199592746.003.0007.

**Soriano C, Valenzuela J. 2009.** Emotion and colour across languages: implicit associations in Spanish colour terms. *Social Science Information* **48**:421–445 DOI 10.1177/0539018409106199.

**Specker E, Leder H, Rosenberg R, Hegelmaier LM, Brinkmann H, Mikuni J, Kawabata H. 2018.** The universal and automatic association between brightness and positivity. *Acta Psychologica* **186**:47–53 DOI 10.1016/j.actpsy.2018.04.007.

**Spence NCW. 1989.** The linguistic field of colour terms in French. *Zeitschrift fur Romanische Philologie* **105**:472–497 DOI 10.1515/zrph.1989.105.5-6.472.

**Strauss ED, Schloss KB, Palmer SE. 2013.** Color preferences change after experience with liked/disliked colored objects. *Psychonomic Bulletin & Review* **20**:935–943 DOI 10.3758/s13423-013-0423-2.

**Sutton TM, Altarriba J. 2016.** Color associations to emotion and emotion-laden words: a collection of norms for stimulus construction and selection. *Behavior Research Methods* **48**:686–728 DOI 10.3758/s13428-015-0598-8.

**Tager A. 2018.** Why was the color violet rarely used by artists before the 1860s?. *Journal of Cognition and Culture* **18**:262–273 DOI 10.1163/15685373-12340030.

**Taylor C, Clifford A, Franklin A. 2013.** Color preferences are not universal. *Journal of Experimental Psychology: General* **142**:1015–1027 DOI 10.1037/a0030273.

**Tham DSY, Sowden PT, Grandison A, Franklin A, Lee AKW, Ng M, Park J, Pang W, Zhao J. 2019.** A systematic investigation of conceptual color associations. *Journal of Experimental Psychology: General* **149(7)**:1311–1332 DOI 10.1037/xge0000703.

**Valdez P, Mehrabian A. 1994.** Effects of color on emotions. *Journal of Experimental Psychology: General* **123**:394–409 DOI 10.1037/0096-3445.123.4.394.

**Wagenmakers E-J. 2007.** A practical solution to the pervasive problems ofp values. *Psychonomic Bulletin & Review* **14**:779–804 DOI 10.3758/BF03194105.

**Wang T, Shu S, Mo L. 2014.** Blue or red? The effects of colour on the emotions of Chinese people. *Asian Journal of Social Psychology* **17**:152–158 DOI 10.1111/ajsp.12050.

**Wang X, Men W, Gao J, Caramazza A, Bi Y. 2020.** Two forms of knowledge representations in the human brain. *Neuron* **107(2)**:383–393 DOI 10.1016/j.neuron.2020.04.010.

**Wexner LB. 1954.** The degree to which colors (hues) are associated with mood-tones. *Journal of Applied Psychology* **38**:432–435 DOI 10.1037/h0062181.

**Winkler S. 2012.** How does color therapy work? *Available at https://health.howstuffworks.com/wellness/spa-health/color-therapy.htm* (accessed on 11 January 2019).

**World Medical Association. 2013.** World Medical Association declaration of Helsinki, ethical principles for medical research involving human subjects. *The Journal of the American Medical Association* **310**:2191–2194 DOI 10.1001/jama.2013.281053.

**Xu J, Dowman M, Griffiths TL. 2013.** Cultural transmission results in convergence towards colour term universals. *Proceedings of the Royal Society B: Biological Sciences* **280**:20123073 DOI 10.1098/rspb.2012.3073.

**Yokosawa K, Schloss KB, Asano M, Palmer SE. 2016.** Ecological effects in cross-cultural differences between U.S. and Japanese color preferences. *Cognitive Science* **40**:1–27 DOI 10.1111/cogs.12291.