# Peer review of "Colour-emotion associations in individuals with red-green colour blindness"

_PeerJ, doi:10.7717/peerj.11180_

## Round 0.1 · original submission · Minor Revisions

I have now received two reviews from experts in the field. I thank the reviewers for their work. As you will read, both reviewers are very positively inclined toward your work, and I share their assessment.
I invite you to revise the text taking the reviews into account. The reviews are thoughtful and clear, so please refer to them for details. In particular, I think that a successful revision will include

a. better grounding in current literature (see comments of both reviewers); I agree with reviewer 1 that you should illustrate the main tests used to evaluate red-green color blindness, both in humans and other animals;

b. I strongly agree with both reviewers that you should take into account previous studies highlighting cultural differences, and clarify further that the results of your study are confined to a Swiss sample;

c. data analysis: reviewer 2 offers excellent suggestions that you may follow.

I am looking forward to reading your revised manuscript.

Reviewer 1 ·

Basic reporting

The aim of the study, which is to evaluate the association between colour and emotions, is absolutely interesting. The manuscript is clear and precise. In the introduction the current knowledge about the topic is well presented and exhaustive. I would recommend describing shortly in the introduction the different tests that can be used to assess red-green colour blindness (rather than only in the supplementary materials). I would also suggest adding also references to the use of behavioural tests to assess vision in the animal kingdom:
- Scholtyssek, C., Kelber, A., & Dehnhardt, G. (2015). Why do seals have cones? Behavioural evidence for colour-blindness in harbour seals. Animal cognition, 18(2), 551-560.
- Siniscalchi, M., d'Ingeo, S., Fornelli, S., & Quaranta, A. (2017). Are dogs red–green colour blind?. Royal Society open science, 4(11), 170869.
- Kelber, A., Vorobyev, M., & Osorio, D. (2003). Animal colour vision–behavioural tests and physiological concepts. Biological Reviews, 78(1), 81-118.

Experimental design

The methodology, as well as the results are clearly described and illustrated. However, light conditions of the testing room should be reported as well as the screen characteristics (refresh rate, resolution, etc.).

Validity of the findings

The discussion is scientifically sound. I appreciated that Authors highlighted the limits (as well as strength) of the study. In my opinion, although recent literature demonstrates that there are similarities across cultures in the association, the influence of the culture of the participants should be taken into account and should be discussed (or eventually considered as a limit of the study). It could be possible, indeed, that social experiences might have produced an association between specific colour and emotional events. For instance, black is usually associated with death in the Western culture.

Reviewer 2 ·

Basic reporting

The authors report an experiment in which they investigated the association between color (with both perceptual and verbal stimuli) and emotional labels in adults with deuteranopia compared to adults without color vision deficiency. Through several data analyses, the authors stated that the color-emotion association is quite similar for both groups for both semantic and perceptual tasks. It is a well-written paper. However, I suggest deepening some points.

Lines 39 to 47: the authors reported evidence about cross-cultural preferences between color and some associated attributes. However, there also some studies that show neither cross-cultural similarities (e.g., Taylor, Clifford & Franklin, 2013; Sorokowski, Sorokowska & Witzel, 2014) nor within a single culture (e.g., Taylor, Schloss, Palmer & Franklin, 2013).

Lines 52 to 56: the authors stated that the presentation mode does not affect the color-emotion association in different cultures. However, they reported a study in which exclusively Swiss people are tested (Jonauskaite et al., 2020b) in both perceptual and semantic tasks. Please soften the sentence so that it is relevant to the findings of Jonauskaite et al. study. In other words, it should be made clear that those data are limited to the Swiss or at least to Western culture.
Furthermore, there is also experimental evidence on language development on color that should be discussed in relation to the universality issue (see for example Ozturk, Shayan, Liszkowski & Majid, 2013).

Overall, I would encourage authors to present and discuss conflicting results as well. The manuscript would benefit from this.

Finally, I was unable to access the data. Please share the data using an international repository (figshare, osf, Mendeley, and so on).

References
Taylor, C., Clifford, A., & Franklin, A. (2013). Color preferences are not universal. Journal of Experimental Psychology: General, 142(4), 1015.

Sorokowski, P., Sorokowska, A., & Witzel, C. (2014). Sex differences in color preferences transcend extreme differences in culture and ecology. Psychonomic bulletin & review, 21(5), 1195-1201.

Taylor, C., Schloss, K., Palmer, S. E., & Franklin, A. (2013). Color preferences in infants and adults are different. Psychonomic bulletin & review, 20(5), 916-922.

Ozturk, O., Shayan, S., Liszkowski, U., & Majid, A. (2013). Language is not necessary for color categories. Developmental Science, 16(1), 111-115.

Experimental design

I appreciate the study design, but some detailed specifications are needed in the method section.

Line 150: Authors should report the number of participants for each condition within the text.
Line 170: Authors should include table S 2 in the text.

Color blindness index: I wonder why you used PCA instead of factorial analysis. PCA is best suited for exploratory data analysis. In your case, I believe you want to detect the latent construct of "color blindness" and factorial analysis should be preferred.

Data analyses

Overall, I appreciate how the data analyses were conducted but found it difficult to follow through as there are too many references to supplemental materials (and maybe the reader too). I suggest including the most relevant tables in the main text and moving some sections away from the main text to supplementary materials (e.g., Color naming). Additionally, the article would benefit from a more detailed graphical representation of the results.

I would recommend the authors to consider re-running the analysis with the multilevel logistic regression model instead of the logistic regression and Bayesian model, including both participants

Validity of the findings

As noted above, the authors should address the issue of conflicting findings on cross-cultural evidence. Furthermore, I found the strength points of the research very clear, but the limitations are not well discussed.

---

## Round 0.2 · Minor Revisions

The paper is very close to acceptance; I invite you to address the last comment of the reviewer, providing a very short overview of the methods used to investigate red-green color blindness in the animals.

Reviewer 1 ·

Basic reporting

The present manuscript has been improved according to the main comments provided by the reviewers and the Editors. However, as suggested before, I would recommend to add some references and literature on the behavioural tests used to evaluate red-green color blindness in the animal kingdom. I understand the reasons behind the authors decision but I strongly believe that a deeper background would significantly improve the manuscript. It would not be necessary to illustrate the tests in details or the reasons why authors preferred the use of a specific test among all the ones described; but I think it could be interesting to report the different methodologies employed, just mentioning them. It could provide a more comprehensive background for the readers.

Experimental design

no comment

Validity of the findings

no comment

---

## Round 0.3 · accepted · Accept

I am happy to inform you that your paper has been accepted for publication on PeerJ.